# Access and use of oxytocin for postpartum haemorrhage prevention: a pre-post study targeting the poorest in six Mesoamerican countries

Aruna M Kamath [ID] ,[1,2] Alexandra M Schaefer,[1] Erin B Palmisano,[1] Casey K Johanns,[1] Alvaro Gonzalez Marmol,[3] Mauricio Dinarte Mendoza,[3] Karla Schwarzbauer,[4] Paola Zúñiga-Brenes,[5] Diego Ríos-Zertuche,[6] Emma Iriarte,[3] Ali H Mokdad,[1] Bernardo Hernandez Prado[1]

For numbered affiliations see end of article.

**Correspondence to**
Dr Aruna M Kamath;
aruna.kamath@seattlechildrens.org

## ABSTRACT

**Objectives** Haemorrhage remains the leading cause of maternal mortality in Central America. The Salud Mesoamérica Initiative aims to reduce such mortality via performance indicators. Our objective was to assess the availability and administration of oxytocin, before and after applying Salud Mesoamérica Initiative interventions in the poorest health facilities across Central America.

**Design** Pre-post study.

**Setting** 166 basic-level and comprehensive-level health facilities in Belize, Guatemala, Honduras, Mexico, Nicaragua and Panama.

**Participants** A random sample of medical records for uncomplicated full-term deliveries (n=2470) per International Classification of Diseases coding at baseline (July 2011 to August 2013) and at first-phase follow-up (January 2014 to October 2014).

**Interventions** A year of intervention implementation prior to first-phase follow-up data collection focused on improving access to oxytocin by strengthening supply chains, procurement, storage practices and pharmacy inventory monitoring, using a results-based financing model.

**Primary and secondary outcome measures** Oxytocin availability (primary outcome) and administration (secondary outcome) for postpartum haemorrhage prevention.

**Results** Availability of oxytocin increased from 82.9% to 97.6%. Oxytocin administration increased from 83.6% to 88.4%. Significant improvements were seen for availability of oxytocin (adjusted OR (aOR)=8.41, 95% CI 1.50 to 47.30). Administration of oxytocin was found to be significantly higher in Honduras (aOR=2.96; 95% CI 1.00 to 8.76) in reference to Guatemala at follow-up.

**Conclusion** After interventions to increase health facility supplies, the study showed a significant improvement in availability but not administration of oxytocin in poor communities within Mesoamerica. Efforts are needed to improve the use of oxytocin.

## INTRODUCTION

Haemorrhage remains the leading cause of maternal mortality both worldwide and in

### Strengths and limitations of this study

► Using a results-based financing model, interventions focused on improving access to oxytocin by strengthening supply chains, procurement, storage practices and pharmacy inventory monitoring,

► Based on a random sampling method of medical records, a multinational data set containing 2470 deliveries was used for the pre-post evaluation of postpartum haemorrhage prevention in the poorest populations of Mesoamerica.

► To measure comparable metrics from varying medical record sources, indicators linked to International Classification of Diseases coding were constructed by the Ministries of Health based on protocols and guidelines of each country.

► We used multivariable logistic regression analysis to evaluate possible factors associated with the primary outcome of oxytocin availability and the secondary outcome of oxytocin administration.

► Challenges to this study involve logistic limitations in collecting data on some of the facility inputs, sample size variation by country and quality of record keeping.

Mesoamerica, accounting for 31.4% of total maternal deaths globally[1] and ranging from 38.0% of total maternal deaths in Nicaragua and Guatemala to 17.9% in Belize based on 2016 Global Burden of Disease Study estimates.[2] While maternal mortality has declined in the last two decades, Millennium Development Goal 5, of reducing maternal mortality by three-fourths, was not achieved globally in 2015.[3 4] A new target has been set for the Sustainable Development Goals to reduce the global maternal mortality ratio to less than 70 per 100 000 live births by 2030.[5] Currently, most of the Mesoamerican countries in this study fall short of or near this target, with the

highest maternal mortality ratio of 109.6 per 100 000 live births occurring in Honduras.[6]

International guidelines for postpartum haemorrhage from the WHO, International Federation of Gynecology and Obstetrics and the International Confederation of Midwives recommend active management of third stage of labour (AMTSL) with emphasis on administering oxytocin, the drug of choice for prevention and treatment of uterine atony.[7 8] Oxytocin is the cornerstone of AMTSL due to its efficacy in reducing postpartum haemorrhage risk by 40%–60%.[9–11] Despite such evidence, translating these best practices into a real-world clinical standard of care is not optimally applied in Mesoamerica, with the proper use of uterotonics as low as 10%–20% in some studies.[12 13]

With efforts to close the evidence-to-practice gap, the Salud Mesoamérica Initiative (SMI) aims to reduce maternal and child mortality for the poorest quintile in Mesoamerica.[14] SMI is a public–private partnership between the Ministries of Health in participating Mesoamerican countries and donors that focuses on improving four major domains of maternal and child health: preventive child health and vaccines; family planning; antenatal care and postpartum care; and essential obstetric and newborn care (EONC).[15] SMI follows a results-based financing model[16 17] in which Ministries of Health commit to achieve negotiated targets for 8–12 performance indicators. Within the EONC domain, health facility performance indicators focused on the availability of equipment (ie, resuscitation equipment, caesarean section kits, stethoscopes), essential medications (ie, antibiotics, antihypertensives) and laboratory inputs (ie, glucometer, cell counter). To receive the performance payment incentive, countries need to achieve at least 80% of targets. At the beginning of each phase, donors contribute approximately half the funding, with the rest from domestic sources.

The first phase of SMI interventions, presented in this study, focused on system readiness by increasing availability of inputs and improving norms. The second upcoming phase directs efforts towards coverage and quality of care. Performance indicators for postpartum haemorrhage prevention involve increasing availability of oxytocin in this first phase and ensuring administration of oxytocin post partum in the second phase.

In this analysis, we examine foremost the availability, and to a lesser extent the administration, of oxytocin for the prevention of postpartum haemorrhage at the first follow-up of the SMI intervention. As such, we assess the extent to which international guidelines are adhered to on the patient level for one aspect of essential obstetric care in poor Mesoamerica.

## METHODS
### Study setting and design
As part of the SMI, 2470 uncomplicated deliveries from 166 health facilities were included in this analysis to assess the availability and administration of oxytocin for the prevention of postpartum haemorrhage in the poorest Mesoamerican communities. These poorest communities were identified by the Initiative administrators based on census data, using as criteria to have the highest concentration of population in the lowest quintile of income. Data collection was conducted at the baseline (July 2011 to August 2013) and the first-phase follow-up (January 2014 to October 2014) time periods.

The first phase, a year of intervention implementation prior to follow-up data collection, focused on supply-side performance indicators. For postpartum haemorrhage prevention, the performance indicator involved increasing the availability of oxytocin, through interventions such as strengthening supply chains (ie, develop efficient distribution routes; optimise supply patterns and frequency of stocking facilities); improving procurement processes (ie, review stock estimates for stock-outs and emergency supplies; establish hospital policies to purchase life-saving medicines), warehouse and pharmacy storage practices (ie, monitor continuously temperature and electricity for cold chains); traffic light systems to monitor expiry dates (ie, colour code expiration dates on a monthly basis); and inventory management processes (ie, first-in first-out utilisation of medicines; and anticipate emergency stock with each reorder cycle).

Facilities were classified into different levels of essential obstetric and neonatal care (EONC)[15]: basic-level facilities manage uncomplicated vaginal deliveries and stabilise patients with complications prior to transfer to higher level of care; and comprehensive-level facilities oversee uncomplicated and complicated births, accept referral patients and perform surgical and emergency care. While the project and its evaluation includes all countries in Mesoamerica, due to availability of data for this specific analysis we used data from Ministry of Health facilities serving the poorest areas in Belize, Guatemala, Honduras, Mexico (state of Chiapas), Nicaragua and Panama. Basic and comprehensive-level health facilities were selected with certainty due to a small number of these hospitals.

A three-part health facility survey was administered at these facilities for both rounds of data collection. The survey included an interview questionnaire to the facility directors on the facility infrastructure and resources; an observation checklist of pharmaceutical inventory and medical and laboratory equipment; and a review of medical records to examine care practices. Medical records of uncomplicated deliveries per International Classification of Diseases (ICD) coding during the specified time frames were sampled at random. Following a systematic sampling method, records were extracted until the required quota for each facility level was met.

With the list of deliveries attended in each facility available, a random starting point was selected in time over the period of study (about 18 months). Cases were then selected with an interval equivalent to the sampling fraction. Sample size of records to be reviewed was calculated due to availability of resources, varying by round,

facility and country. Therefore, a larger overall sample size was assigned to countries with a larger operation (Honduras, Guatemala and Mexico), and was distributed across the facilities to be surveyed depending on their EONC level. Expected sample sizes had enough power to detect differences in evaluation indicators, including treatment to deliveries according to the national norms. Due to availability of resources, it was possible to increase the number of cases in the sample in the follow-up to increase power, especially in the countries with a reduced number of cases in the baseline (Belize). Overall, we had a power over 80% to detect differences of 10 percentage points between baseline and follow-up for oxytocin availability in all countries except Belize and Nicaragua. Additional information of the methodology is available elsewhere.[14 17–19]

To measure standardised, replicable and comparable metrics from varying medical record sources, indicators that were linked to discharge diagnosis and ICD coding were constructed by the Ministries of Health based on the protocols and guidelines of each country. Criteria checklists for each indicator differed by EONC facility level. These criteria checklists were then transformed into data points and conditional algorithms for data collection. During field visits, medical record data availability and measurability of criteria were assessed.[20] Data collection was conducted by trained physicians and nurses from the region, and data were submitted electronically using the survey software DatStat Illume.

Availability of oxytocin was defined as supply on the day of survey visit; and administration of oxytocin was defined as given intravenously or intramuscularly for postpartum haemorrhage prevention (excluding postpartum haemorrhage treatment). Uncomplicated delivery records selected for review included full-term (>37 weeks) gestational age at participating health facilities, including caesarean sections without complications per ICD coding. Deliveries with adverse outcomes per ICD coding that required haemorrhage treatment were excluded, due to collection under a separate survey module.

## Statistical analysis

We used multivariable logistic regression analysis to evaluate possible facility-level factors associated with the primary outcome of oxytocin availability and the secondary outcome of oxytocin administration. Covariates evaluated in this study were timing of data collection (baseline vs first-phase follow-up), country and EONC facility type (basic vs comprehensive). Additionally for oxytocin administration, we examined relevant training within the last year (routine labour care, basic emergency obstetrical care or maternal complications care); and oxytocin availability (day of survey visit), and skilled personnel (doctor, nurse). When fitting the regression models, skilled personnel was dropped from the model due to predicting success perfectly, meaning a skilled personnel was always present for the administration of oxytocin. The rate of missing data was 4.4% for oxytocin

availability and 0.7% for oxytocin administration for the regression analysis. P values <0.05 were considered significant. We used Stata V.14.2 (StataCorp, College Station, TX, USA) for the analysis.

## Patient and public involvement

Patients or the public were not involved in the design, conduct, reporting or dissemination of our research.

## RESULTS

### Sample characteristics

Table 1 presents the medical records, patient, facility and personnel characteristics. The majority of deliveries were in basic-level health facilities (64.3%). We sampled the most medical records from Guatemala (27.5%), followed by Chiapas (Mexico) (20.8%), Honduras (19.6%), Panama (16.0%), Nicaragua (11.9%) and Belize (4.2%), based on the quota for records within each facility, prevalence of records of interest and sampling interval. With a range from 11 to 48 years, the mean maternal age was 24.3 years. Of these patients, 55.2% were single (no social partnership), and 72.7% attained less than a secondary education. Nearly all patients (99.7%) had a single gestational pregnancy, and only 0.3% had a multiple gestational pregnancy. Most of these patients delivered vaginally (97.7%) as opposed to by caesarean section (2.3%). Supply (on the day of survey visit) for oxytocin at baseline was 82.9%, which increased to near complete availability (97.6%) at first-phase follow-up. Personnel on staff—that is, skilled birth attendants employed by the facility—at baseline was 96.2% for physicians and 91.0% for nurses and increased to 100% and 93.2%, respectively, at follow-up. Relevant training—that is, instruction on routine labour care, basic emergency obstetrical care or maternal complications care—provided within the last year increased from 74.0% at baseline to 92.0% at first-phase follow-up.

### Availability and administration of oxytocin for postpartum haemorrhage prevention

Availability of oxytocin (on the day of the survey) increased from 82.9% to 97.6% and administration of oxytocin increased from 83.6% to 88.4% from baseline to first-phase follow-up. Notably, all countries achieved 100% availability of oxytocin at first-phase follow-up, except for Guatemala (91.7). Comprehensive-level facilities achieved 100% availability, compared with basic-level facilities (96.8%) at first-phase follow-up. Administration of oxytocin at first-phase follow-up ranged from 80.0% in Belize to 94.8% in Honduras. Oxytocin administration was nearly the same at basic (88.4%) and comprehensive-level (88.5%) facilities at first-phase follow-up (table 2).

Table 3 shows possible first-phase follow-up and facility determinants of the availability (primary outcome) and administration of oxytocin (secondary outcome). Odds of availability were significantly higher at first-phase follow-up compared with baseline (adjusted OR

**Table 1** Medical records, patient and facility characteristics by first-phase follow-up

| | n (%) | | |
|---|---|---|---|
| | **Baseline 922 (37.3)** | **Follow-up 1548 (62.7)** | **Total 2470* (100)** |
| **Medical record characteristics** | | | |
| EONC facility type | | | |
| Basic level | 616 (66.8) | 971 (62.7) | 1587 (64.3) |
| Comprehensive level | 306 (33.2) | 577 (37.3) | 883 (35.8) |
| Country | | | |
| Belize | 14 (1.5) | 90 (5.8) | 104 (4.2) |
| Guatemala | 247 (26.8) | 432 (27.9) | 679 (27.5) |
| Honduras | 234 (25.4) | 249 (16.1) | 483 (19.6) |
| Mexico | 180 (19.5) | 334 (21.6) | 514 (20.8) |
| Nicaragua | 90 (9.8) | 205 (13.2) | 295 (11.9) |
| Panama | 157 (17.0) | 238 (15.4) | 395 (16.0) |
| **Patient characteristics** | | | |
| Maternal age† | 24.3 (6.7) | 24.3 (6.6) | 24.3 (6.6) |
| Marital status | | | |
| Single | 409 (61.8) | 656 (51.8) | 1065 (55.2) |
| Partnership | 253 (38.2) | 610 (48.1) | 863 (44.8) |
| Education | | | |
| Less than secondary | 392 (70.5) | 772 (73.9) | 1164 (72.7) |
| Secondary or higher | 164 (29.5) | 273 (26.1) | 437 (27.3) |
| Pregnancy type | | | |
| Single gestational | 154 (98.7) | 917 (99.9) | 1071 (99.7) |
| Multiple gestational | 2 (1.3) | 1 (0.1) | 3 (0.3) |
| Delivery type | | | |
| Vaginal | 409 (98.6) | 1091 (97.3) | 1500 (97.7) |
| Caesarean section | 6 (1.5) | 30 (2.7) | 36 (2.3) |

| | n (%) | | |
|---|---|---|---|
| | **Baseline 78 (47.0)** | **Follow-up 88 (53.0)** | **Total 166 (100)** |
| **Facility and personnel characteristics** | | | |
| Oxytocin supply‡ | 58 (82.9) | 83 (97.6) | 141 (91.0) |
| Personnel on staff§ | | | |
| Physician | 75 (96.2) | 88 (100) | 163 (98.2) |
| Nurse | 71 (91.0) | 82 (93.2) | 153 (92.2) |
| Relevant training¶ | 57 (74.0) | 81 (92.0) | 138 (83.6) |

*n may vary for each variable due to missingness.
†Mean±SD.
‡Day of survey visit.
§Skilled birth attendants employed by the facility. Midwife excluded due to varying skill levels among countries.
¶Within last year, includes routine labour care, basic emergency obstetrical care and maternal complications.
EONC, essential obstetric and newborn care.

(aOR)=8.41, 95 % CI 1.50 to 47.30), while odds of administration increased in magnitude but were not significant at first-phase follow-up (aOR=1.63, 95 % CI 0.83 to 3.21).

**Table 2** Oxytocin availability and administration by facility type, country and first-phase follow-up

| | n (%) | | |
|---|---|---|---|
| **Oxytocin availability** | **Baseline 58 (82.9)** | **Follow-up 83 (97.6)** | **Total* 141 (91.0)** |
| EONC facility type | | | |
| Basic level | 43 (84.3) | 61 (96.8) | 104 (91.2) |
| Comprehensive level | 15 (78.9) | 22 (100.0) | 37 (90.2) |
| Country | | | |
| Belize | 4 (100.0) | 4 (100.0) | 8 (100.0) |
| Guatemala | 15 (88.2) | 22 (91.7) | 37 (90.2) |
| Honduras | 13 (92.9) | 12 (100.0) | 25 (96.2) |
| Mexico | 7 (100.0) | 14 (100.0) | 21 (75.0) |
| Nicaragua | 6 (85.7) | 14 (100.0) | 20 (95.2) |
| Panama | 13 (92.9) | 17 (100.0) | 30 (96.8) |
| **Oxytocin administration for prevention of postpartum haemorrhage** | **n (%)** | | |
| | **Baseline 771 (83.6)** | **Follow-up 1353 (88.4)** | **Total 2124 (86.6)** |
| EONC facility type | | | |
| Basic level | 521 (84.6) | 852 (88.4) | 1373 (86.9) |
| Comprehensive level | 250 (81.7) | 501 (88.5) | 751 (86.1) |
| Country | | | |
| Belize | 9 (64.3) | 72 (80.0) | 81 (77.9) |
| Guatemala | 198 (80.2) | 393 (91.0) | 591 (87.0) |
| Honduras | 224 (95.7) | 221 (94.8) | 445 (95.3) |
| Mexico | 129 (71.7) | 277 (82.9) | 406 (79.0) |
| Nicaragua | 87 (96.7) | 168 (82.8) | 255 (87.0) |
| Panama | 124 (79.0) | 222 (93.3) | 346 (87.6) |

*n may vary for each variable due to missingness.
EONC, essential obstetric and newborn care.

Additionally, administration of oxytocin was found to be significantly higher in Honduras (aOR=2.96; 95 % CI 1.00 to 8.76) in reference to Guatemala. Other covariates examined, such as comprehensive-level facility, relevant training and oxytocin availability, showed a positive but not significant correlation to the outcomes.

## DISCUSSION

There remains a paucity of studies with regard to oxytocin access and use in Mesoamerica[13 21–23] or in low-resource countries in general.[12 24–30] In Mesoamerica, García-Elorrio et al [13] and Low et al[21] conducted pre-post evaluations at health facilities within a single country, rather than a multinational analysis. These studies focused on oxytocin administration interventions, such as multifaceted provider skills in Nicaragua and AMTSL training in Honduras, respectively, without attention to improving oxytocin availability, the primary outcome of this study. Therefore, we provide an updated, regional evaluation

**Table 3** Factors associated with availability (primary outcome) and administration of oxytocin (secondary outcome)

| | Oxytocin availability | | Oxytocin administration for prevention of postpartum haemorrhage | |
| --- | --- | --- | --- | --- |
| | Crude OR (95% CI) (n=2257) | Adjusted OR* (95% CI) (n=2257) | Crude OR (95% CI) (n=2343) | Adjusted OR (95% CI) (n=2343) |
| First-phase follow-up | **7.29 (1.48 to 35.84)** | **8.41 (1.50 to 47.30)** | 1.43 (0.79 to 2.62) | 1.63 (0.83 to 3.21) |
| Country | | | | |
| Belize | Predicts success perfectly† | Predicts success perfectly | 0.52 (0.13 to 2.12) | 0.44 (0.11 to 1.71) |
| Guatemala | 1 | 1 | 1 | 1 |
| Honduras | 1.70 (0.18 to 16.46) | 2.24 (0.27 to 18.24) | **2.71 (1.02 to 7.22)** | **2.96 (1.00 to 8.76)** |
| Mexico | 0.67 (0.14 to 3.10) | 0.51 (0.11 to 2.46) | 0.57 (0.25 to 1.31) | 0.57 (0.24 to 1.37) |
| Nicaragua | 2.22 (0.23 to 21.71) | 2.01 (0.19 to 20.92) | 1.00 (0.38 to 2.62) | 0.95 (0.35 to 2.54) |
| Panama | 3.43 (0.36 to 33.05) | 3.83 (0.36 to 41.22) | 1.04 (0.41 to 2.66) | 1.03 (0.40 to 2.61) |
| Comprehensive-level facility | 1.00 (0.24 to 4.15) | 1.29 (0.31 to 5.31) | 0.92 (0.47 to 1.81) | 0.96 (0.46 to 2.00) |
| Relevant training‡ | – | – | 1.49 (0.71 to 3.13) | 1.08 (0.44 to 2.67) |
| Oxytocin availability§ | – | – | 2.01 (0.92 to 4.39) | 1.45 (0.66 to 3.19) |

All models clustered at health facility level.

*Adjusted for first-phase follow-up, country and comprehensive-level facility. Relevant training and oxytocin availability not included (not applicable).

†Belize attained availability of oxytocin at all times (regression for this binary variable not possible).

‡Within last year, includes routine labour care, basic emergency obstetrical care and maternal complications.

§Day of survey visit.

of postpartum haemorrhage prevention practices that may guide clinical management and policy priorities for this targeted population. Our study showed that the availability of oxytocin improved significantly after system readiness interventions, among the poorest population within Mesoamerica.

This analysis attests that clinical standards of care could be achieved in Mesoamerica with proper monitoring and management of oxytocin supplies. As expected from the implementation timeline, the first stage of the SMI focused on improving system readiness, namely strengthening facility infrastructure, supply chains, procurement processes, storage practices and inventory monitoring of drugs and equipment.[14 16] Consequently, increasing the availability of oxytocin, the primary goal of the first phase of the initiative, was achieved. Administration of oxytocin, a lesser goal, increased but not in the same proportion as availability, which suggests that increasing the availability of inputs is necessary but not sufficient to improve quality of care. Comprehensive quality improvement strategies would be needed that involve increasing availability of equipment and supplies, and improving skills

and abilities of personnel, and establishing mechanisms to improve processes, identify causes and lift barriers to ensure proper practices of care in the next phase.

This study has some limitations. First, oxytocin availability was defined based on the day of the visit to health facilities, which may not match the day of delivery in a medical record. Due to logistic limitations, it was not possible to assess stock-outs of oxytocin over a longer period of time. Therefore, we must consider oxytocin availability on the day of the visit as a proxy for availability over a longer period of time. Quality of available oxytocin cannot be included when defining oxytocin availability, as data were not collected on expiration or storage of oxytocin. Second, this study focused on uncomplicated full-term deliveries and did not include patients resulting in haemorrhage due to two separate survey modules for data collection. In the module with adverse outcomes from haemorrhage, uterotonics were often administered more than once without distinction of whether for prevention or treatment. Third, sample sizes also vary by country, and in some cases like Belize, the reduced sample size limited our capacity to have country-specific precise

estimates and to detect significant differences between baseline and follow-up measurements. Fourth, quality of record keeping may affect the information used in this study and may vary by country. Nonetheless, we used a standard methodology across countries, automated data collection and used quality control measures that ensure comparability between countries.

## CONCLUSION

The study showed a significant improvement in availability of oxytocin at facilities that provide care to poor communities within Mesoamerica, but not optimal administration, as expected after emphasis on supply-side interventions. Continued monitoring and evaluation, beyond input availability, are essential to better understand how to improve processes and clinical practices. Our study provides valuable information to close the evidence-to-practice gap and to reduce maternal mortality in Mesoamerica.

**Author affiliations**
[1]Department of Health Metrics Sciences, Institute for Health Metrics and Evaluation, Seattle, Washington, USA
[2]Department of Anesthesiology and Pain Medicine, University of Washington, Seattle, Washington, USA
[3]Salud Mesoamerica Initiative, Inter-American Development Bank, Panama City, Panama
[4]Salud Mesoamerica Initiative, Inter-American Development Bank, Tegucigalpa, Honduras
[5]Salud Mesoamerica Initiative, Inter-American Development Bank, San José, Costa Rica
[6]Salud Mesoamerica Initiative, Inter-American Development Bank, Washington, DC, USA

**Acknowledgements** The authors recognise our colleagues at the Inter-American Development Bank, the Institute for Health Metrics and Evaluation and our collaborating data collection institutions across Mesoamerica: University of Belize in Belize, Fundación FES in Guatemala and Honduras, El Colegio de la Frontera Sur in Mexico and Nicaragua, Centro de Investigación y Estudios en Salud/Universidad de Nicaragua in Nicaragua and UNIMER Centroamerica in Guatemala, Mexico, Honduras, Nicaragua, Panama and Belize. The authors also acknowledge Adrienne Chew from the Institute for Health Metrics and Evaluations for editing this manuscript.

**Contributors** AMK conceived the study and prepared data analysis, interpretation of data, initial draft and final manuscript. AMS, EBP and CKJ contributed to and reviewed data analysis. AGM, MDM, KS, PZB, DRZ and EI contributed to the acquisition of the data. BHP and AHM supervised the study and data compilation and contributed to interpretation of data. All authors critically reviewed the draft and approved the final manuscript.

**Funding** Salud Mesoamérica Initiative (SMI) is a public–private partnership funded by the Bill & Melinda Gates Foundation (grant number OPPGH5328), Agencia Española de Cooperación Internacional para el Desarrollo (10.13039/501100004892) and the Carlos Slim Foundation (10.13039/100004429), through the Inter-American Development Bank. AMK is funded by the National Institutes of Health T32 grant (T32GM086270).

**Disclaimer** The funders had no role in study design, data collection, data analysis and interpretation, or preparation of the manuscript.

**Competing interests** None declared.

**Patient and public involvement** Patients and/or the public were not involved in the design, or conduct, or reporting, or dissemination plans of this research.

**Patient consent for publication** Not required.

**Ethics approval** Approval for this study involved the institutional review board from the University of Washington (exemption as a non-human subject research

determination), partnering data collection agencies (El Colegio de la Frontera Sur-Mexico), the Ministry of Health in each country (Belize, Guatemala, Honduras, Mexico, Nicaragua, Panama) and the indigenous communities in Panama and Mexico. Prior to data collection, informed consent was obtained from each health facility administrator. During the data extraction process, medical records were anonymised. No patient consent was required.

**Provenance and peer review** Not commissioned; externally peer reviewed.

**Data availability statement** Datasets for Salud Mesoamerica Initiative baseline and follow-up surveys used in this study are publicly available at the Global Health Data Exchange (GHDx) website: http://ghdx.healthdata.org/

**ORCID iD**
Aruna M Kamath http://orcid.org/0000-0002-7104-0996

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
