## [Reviewer comments · BMJ Open]

ARTICLE DETAILS

TITLE (PROVISIONAL)	Access and use of oxytocin for post-partum hemorrhage prevention: a pre-post study targeting the poorest in six Mesoamerican countries
AUTHORS	Kamath, Aruna; Schaefer, Alexandra; Palmisano, Erin; Johanns, Casey; Gonzalez Marmol, Alvaro; Dinarte Mendoza, Mauricio; Schwarzbauer, Karla; Zúñiga-Brenes, Paola; Ríos-Zertuche, Diego; Iriarte, Emma; mokdad, ali; Hernandez Prado, Bernardo

VERSION 1 - REVIEW

REVIEWER	Dr Belemsaga/Yugbaré Danielle Institut de recherche en sciences de la santé (IRSS)
REVIEW RETURNED	04-Oct-2019

GENERAL COMMENTS	The paper titled “Access and use of oxytocin for post-partum hemorrhage prevention: targeting the poorest in six Mesoamerican countries » aimed to analyse oxytocin use among the poorest population within Mesoamerica countries. The article's findings on the effects of the Salud Mesoamerica initiative on the higher access and use of oxytocin contribute to improve maternal health. The manuscript should be easily read without some repetition. The main comments are on the introduction and the discussion. The discussion need to be revised. Abstract I would suggest describing briefly the Salud Mesoamerica initiative in the abstract background. Strengths and limitations I would suggest to delete this subtitle: add the strengths in the introduction and the limitations at the end of the discussion if it is not already in. Introduction “based on recent estimates.(2)” specify the author and the year in the text. I would suggest for all references (when require) to put the reference before the stop “(2).” Methods “Quality of data in medical records varied by facility and country”. I would suggest to delete or to move it to the limitations at the end of the discussion if its not already in. Line 149-153 « Approval for this study....medical records were anonymized ». move to Ethics approval line 284 Discussion
---

	I would suggest deleting the subtitles of the discussion (main results, interpretation...) Line 231 to 240 "There remains...the most underserved populations" This paragraph seems to be a justification of the study than a discussion or interpretation of the results. I would suggest to keep the first sentence can be kept in the discussion, move line 232-240 to the introduction and overall reorganize the introduction. I would suggest discussing the single country studies with yours (methods, results, comparability...).
--	--

REVIEWER	Joshua D. Dahlke MD Nebraska Methodist Women's Hospital and Perinatal Center, Omaha, Nebraska, USA
REVIEW RETURNED	07-Nov-2019

GENERAL COMMENTS	In the article under consideration, the authors performed a retrospective cohort study of oxytocin availability and administration after the Salud Mesoamerica Initiative intervention on postpartum hemorrhage prevention. Specific comments to follow:  1. In general, the manuscript represents an analysis of a very important subject with multiple logistical obstacles. 2. It is important to consistently distinguish oxytocin availability and oxytocin usage throughout the manuscript with clear distinction of which is the primary outcome for this study. (For example, in the abstract, the results of usage is mentioned before availability) My interpretation is that the first phase of the SMI and the primary outcome of interest in this study is oxytocin availability. It may be beneficial to the reader to emphasize these findings more than the usage aspect of their findings. 3. With that in mind, what was the specific interventions accounted for increased oxytocin availability? The authors describe strengthening supply chains, improving procurement, storage practices, expiration monitoring, inventory management etc as interventions to achieve this goal. It may be beneficial to the reader to further describe these aspects more specifically. 4. Consider including adding months to the time periods (baseline and first-phase) to specify periods of interest. Presumably these are calendar year time periods. 5. It may be beneficial to the reader to briefly describe the 8-12 other performance indicators in the SMI other than postpartum hemorrhage. This may allow the reader to appreciate the depth and breadth of such an undertaking. 6. In the Methods section, the authors describe how data was obtained (systematic sampling, etc). In this section, it is stated that the 'size of records to be reviewed was calculated due to availability of resources.' While they cite other studies from the SMI, it may be beneficial to the reader to clarify a bit more what is meant by this statement. 7. Regarding oxytocin administration, if available, it would be interesting to know the proportion who received IV vs IM Oxytocin 8. In the Main Findings section of the Discussion, the authors start with a statement regarding oxytocin use. Consider reframing to emphasize Oxytocin administration as this seems to be the primary outcome of interest in this phase of the initiative. I appreciate the opportunity to review this manuscript.
--

VERSION 1 – AUTHOR RESPONSE

Reviewer 1:

Comment 1:

The article's findings on the effects of the Salud Mesoamerica initiative on the higher access and use of oxytocin contribute to improve maternal health.

The manuscript should be easily read without some repetition.

Response 1:

We thank the reviewer for this overall comment.

Throughout the manuscript, we have edited for readability and repetition. Please refer to Reviewer 1 Response 7, Reviewer 1 Response 8, Reviewer 1 Response 10, Reviewer 2 Response 2.

Comment 2:

The main comments are on the introduction and the discussion. The discussion need to be revised.

Response 2:

Introduction and discussion sections have been revised. Please refer edits in Reviewer 1 Comment 5, Reviewer 1 Comment 10 and Reviewer 1 Comment 11.

Comment 3:

Abstract - I would suggest describing briefly the Salud Mesoamerica initiative in the abstract background.

Response 3:

We would like to clarify that the Abstract was reformatted per journal instructions to include an objective section, rather than a background section. However, as suggested by the Reviewer, a brief description of Salud Mesoamerica Initiative has been added in the objectives section, on p.2, as follows:

“Hemorrhage remains the leading cause of maternal mortality in Central America. The Salud Mesoamerica Initiative aims to reduce such mortality via performance indicators.”

Comment 4:

Strengths and limitations

I would suggest to delete this subtitle: add the strengths in the introduction and the limitations at the end of the discussion if it is not already in.

Response 4:

We would like to clarify that per journal instructions the “Strengths and limitations of this study” section is required, and therefore we are unable to delete this section as recommended. However, as suggested by the Reviewer, the content of this section also exists in the Introduction and limitations section of the Discussion.

Comment 5:

Introduction

“based on recent estimates.(2)” specify the author and the year in the text.

Response 5:

This sentence has been revised, on p.3, as follows:

“Hemorrhage remains the leading cause of maternal mortality both worldwide and in Mesoamerica, accounting for 31.4% of total maternal deaths globally(1) and ranging from 38.0% of total maternal deaths in Nicaragua and Guatemala to 17.9% in Belize based on 2016 Global Burden of Disease Study estimates.(2)”

Comment 6:

Introduction

I would suggest for all references (when require) to put the reference before the stop “(2).”

Response 6:

We would like to clarify that the reference style used in the manuscript reflects BMJ Open’s latest published issue.

Comment 7:

Methods

“Quality of data in medical records varied by facility and country”. I would suggest to delete or to move it to the limitations at the end of the discussion if its not already in.

Response 7:

We agree with this repetition. This sentence has been deleted, as it has been addressed in the Limitations section, on p.14, as follows:

“Fourth, quality of record keeping may affect the information used in this study and may vary by country. Nonetheless, we used a standard methodology across countries, automated data collection, and used quality control measures that ensure comparability between countries.”

Comment 8:

Methods

Line 149-153 « Approval for this study... medical records were anonymized ». move to Ethics approval line 284

Response 8:

This paragraph has been moved to the Ethics approval section on p.15.

Comment 9:

Discussion

I would suggest deleting the subtitles of the discussion (main results, interpretation...)

Response 9:

Subtitles have been deleted from the Discussion section.

Comment 10:

Discussion

Line 231 to 240 "There remains...the most underserved populations" This paragraph seems to be a justification of the study than a discussion or interpretation of the results. I would suggest to keep the first sentence can be kept in the discussion, move line 232-240 to the introduction and overall reorganize the introduction.

Response 10:

We agree with this repetition in the Discussion section (the rest of the paragraph is summarized sufficiently by the first sentence with same cited references). Thus, we kept the first sentence of this paragraph in the discussion section and deleted the rest of the paragraph. We did not move the rest of the paragraph to the Introduction and reorganize the Introduction for the following reason:

As the Introduction section already states, on p.4: "Oxytocin is the cornerstone of AMTSL due to its efficacy in reducing postpartum hemorrhage risk by 40–60%.(9–11) Despite such evidence, translating these best practices into a real-world clinical standard of care is not optimally applied in Mesoamerica, with the proper use of uterotonics as low as 10–20% in some studies.(12,13)," we feel that these additional sentences in the Introduction would not add value to the manuscript, but rather repetition.

Comment 11:

Discussion

I would suggest discussing the single country studies with yours (methods, results, comparability...).

Response 11:

Comparison between single country studies and this analysis have been added to the Discussion section, on p. 12, as follows:

“In Mesoamerica, García-Elorrio et al 2014 and Low et al 2012 conducted pre-post evaluations at health facilities within a single country, rather than a multinational analysis. These studies focused on oxytocin administration interventions, such as multifaceted provider skills in Nicaragua and AMTSL training in Honduras respectively, without attention to improving oxytocin availability, the primary outcome of this study.”

Reviewer 2:

Comment 1:

In general, the manuscript represents an analysis of a very important subject with multiple logistical obstacles.

Response 1:

We thank the reviewer for this overall comment.

Comment 2:

It is important to consistently distinguish oxytocin availability and oxytocin usage throughout the manuscript with clear distinction of which is the primary outcome for this study. (For example, in the abstract, the results of usage is mentioned before availability) My interpretation is that the first phase of the SMI and the primary outcome of interest in this study is oxytocin availability. It may be beneficial to the reader to emphasize these findings more than the usage aspect of their findings.

Response 2:

We have ensured that oxytocin availability and oxytocin usage is consistently distinguished throughout the manuscript, with emphasis on the primary outcome.

Abstract section has been reformatted, with primary and secondary outcomes, on p.2, as follows:

“Primary and secondary outcome measures: Oxytocin availability (primary outcome) and administration (secondary outcome) for post-partum hemorrhage prevention.”

In the Introduction section, this distinction has been noted, on p.5, as follows:

“The first phase of SMI interventions, presented in this study, aims to improve system readiness by increasing availability of inputs and improving norms. The second upcoming phase targets improvements in coverage and quality of care. Specifically, performance indicators for postpartum hemorrhage prevention involve increasing availability of oxytocin in this first phase and ensuring administration of oxytocin postpartum in the second phase.”

“In this analysis, a multinational analysis of oxytocin use for this targeted population and region, we examine foremost the availability, and to a lesser extent administration, of oxytocin for the prevention of postpartum hemorrhage at the first follow-up of the SMI intervention.”

In the Methods section, on p. 5, the first phase of the study is emphasized (oxytocin availability), with no discussion on the second phase (oxytocin administration).

Primary and secondary outcomes are identified in the statistical analysis section of the Methods section, on p.8, as follows:

“We used multivariable logistic regression analysis to evaluate possible facility-level factors associated with the primary outcome of oxytocin availability and secondary outcome of oxytocin administration.”

In the Results section, order of describing results prioritizes oxytocin availability results first, followed by oxytocin administration results. Table 3 has been updated to designate primary and secondary outcomes clearly.

In the Discussion section, first paragraph on p.12-13, lesser findings on oxytocin administration have been deleted, now highlighting only the significant findings of oxytocin availability from the first phase.

In the Discussion section, next paragraph on p.13, first stage is emphasized of the implementation timelines, while the second stage has been deleted.

Also, in the discussion section, primary outcome versus secondary outcome have been delineated, on p.13, as follows:

“Consequently, increasing the availability of oxytocin, the primary goal of the first phase of the initiative, was achieved. While, administration of oxytocin, a lesser goal, increased but not in the same proportion as availability, which suggests that increasing the availability of inputs is necessary but not sufficient to improve quality of care.”

Comment 3:

With that in mind, what were the specific interventions accounted for increased oxytocin availability? The authors describe strengthening supply chains, improving procurement, storage practices, expiration monitoring, inventory management etc as interventions to achieve this goal. It may be beneficial to the reader to further describe these aspects more specifically.

Response 3:

In the Methods section, we provide specific examples of the interventions described, on p.5-6, as follows:

“For postpartum hemorrhage prevention, the performance indicator involved increasing the availability of oxytocin, through interventions such as strengthening supply chains (i.e. develop efficient distribution routes; optimize supply patterns and frequency of stocking facilities); improving procurement processes (i.e. review stock estimates for stock outs and emergency supplies; establish hospital policies to purchase life-saving medicines), warehouse and pharmacy storage practices (i.e. monitor continuously temperature and electricity for cold chains); traffic light systems to monitor expiry dates (i.e. color code expiration dates on a monthly basis); and inventory management processes (i.e. first-in first-out utilization of medicines; anticipate emergency stock with each reorder cycle).”

Comment 4:

Consider including adding months to the time periods (baseline and first-phase) to specify periods of interest. Presumably these are calendar year time periods.

Response 4:

In the Abstract and Methods section, on p.2 and p.5 respectively, data collection time periods have been specified by months, as follows:

“... baseline (July 2011–August 2013) and at first-phase follow-up (January 2014–October 2014).”

Comment 5:

It may be beneficial to the reader to briefly describe the 8-12 other performance indicators in the SMI other than postpartum hemorrhage. This may allow the reader to appreciate the depth and breadth of such an undertaking.

Response 5:

While performance indicators like oxytocin availability spanned several countries, other performance indicators varied by country-led decisions. Thus, we have provided more details on the performance indicators within the EONC domain in the Introduction section, on p.4, as follows:

“SMI is a public-private partnership between the Ministries of Health in participating Mesoamerican countries and donors that focuses on improving four major domains of maternal and child health: preventive child health and vaccines; family planning; antenatal care and postpartum care; and Essential Obstetric and Newborn Care (EONC).⁽¹⁵⁾ SMI follows a results-based financing model^(16,17) in which Ministries of Health commit to achieve negotiated targets for eight to twelve performance indicators. Within the EONC domain, health facility performance indicators focused on the availability of equipment (i.e. resuscitation equipment, cesarean section kits, stethoscopes), essential medications (i.e. antibiotics, antihypertensives), and laboratory inputs (i.e. glucometer, cell counter).”

Comment 6:

In the Methods section, the authors describe how data was obtained (systematic sampling, etc). In this section, it is stated that the ‘size of records to be reviewed was calculated due to availability of resources.’ While they cite other studies from the SMI, it may be beneficial to the reader to clarify a bit more what is meant by this statement.

Response 6:

In the Methods section, we have clarified this sentence, on p.7, as follows:

“Therefore, a larger overall sample size was assigned to countries with a larger operation (Honduras, Guatemala and Mexico), and was distributed across the facilities to be surveyed depending on their EONC level.”

Comment 7:

Regarding oxytocin administration, if available, it would be interesting to know the proportion who received IV vs IM Oxytocin

Response 7:

Route of administration (IV, IM) data was not always captured, and therefore not available to include in this analysis.

Comment 8:

In the Main Findings section of the Discussion, the authors start with a statement regarding oxytocin use. Consider reframing to emphasize Oxytocin administration as this seems to be the primary outcome of interest in this phase of the initiative.

Response 8:

In reorganizing this paragraph, this first sentence has been deleted. However, we have edited this paragraph to emphasize clearly the primary outcome of the study, on p.12, as follows:

"... improving oxytocin availability, the primary outcome of this study."

Also in this paragraph, as mentioned in Reviewer 2 Comment 2, lesser findings on oxytocin administration have been deleted, now highlighting only the significant findings of oxytocin availability from the first phase.

VERSION 2 – REVIEW

REVIEWER	Joshua D. Dahlke MD Nebraska Methodist Hospital and Perinatal Center, Omaha, Nebraska
REVIEW RETURNED	11-Dec-2019

GENERAL COMMENTS	The authors have sufficiently addressed initial review comments. I appreciate the opportunity to review this manuscript
---